# Mandibular Reconstruction with Osseous Free Flap and Immediate Prosthetic Rehabilitation (Jaw-in-a-Day): In-House Manufactured Innovative Modular Stackable Guide System

**DOI:** 10.3390/bioengineering11121254

**Published:** 2024-12-11

**Authors:** Matthias Ureel, Pieter-Jan Boderé, Benjamin Denoiseux, Pasquier Corthouts, Renaat Coopman

**Affiliations:** 1Department of Oral and Craniomaxillofacial Surgery, Ghent University Hospital, 9000 Ghent, Belgium; benjamin.denoiseux@uzgent.be (B.D.); pasquier_corthouts@hotmail.com (P.C.); renaat.coopman@ugent.be (R.C.); 2Department of Dentistry, Ghent University Hospital, 9000 Ghent, Belgium; pieter-jan.bodere@uzgent.be

**Keywords:** virtual surgical planning, 3D printing, additive manufacturing, in-house production, fibula free flap, oncologic reconstruction, dental rehabilitation, jaw-in-a-day

## Abstract

**Background:** Head and neck reconstruction following ablative surgery results in alterations to maxillofacial anatomy and function. These postoperative changes complicate dental rehabilitation. **Methods:** An innovative modular, stackable guide system for immediate dental rehabilitation during mandibular reconstruction is presented. The virtual surgical planning was performed in Materialise Innovation Suite v26 and Blender 3.6 with the Blenderfordental add-on. The surgical guides and models were designed and manufactured at the point of care. **Results:** The duration of the surgery was 9 h and 35 min. Good implant stability (>35 Ncm) and a stable occlusion were achieved. After 9 months of follow-up, the occlusion remained stable, and a mouth opening of 25 mm was registered. The dental implants showed no signs of peri-implant bone loss. Superposition of the preoperative planning and postoperative position of the fibula parts resulted in an average difference of 0.70 mm (range: −1.9 mm; 5.4 mm). **Conclusions:** The in-house developed stackable guide system resulted in a predictive workflow and accurate results. The preoperative virtual surgical planning was time-consuming and required extensive CAD/CAM and surgical expertise. The addition of fully guided implant placement to this stackable guide system would be beneficial. More research with longer follow-ups is necessary to validate these results.

## 1. Introduction

Oral rehabilitation after ablative surgery in the head and neck region is very important for the patient’s functional outcome and quality of life (QOL) [1]. Extensive resections in the head and neck area significantly alter the facial and oral anatomy, often resulting in a shallow vestibule, lip incompetence, sensory disturbances, and trismus. These anatomical changes significantly compromise the prosthetic retention possibilities, complicating rehabilitation with removable prostheses [2].

The use of dental implants as support for dental prostheses has facilitated oral rehabilitation in oral cancer patients. Primary implantation was an important first evolution [3,4]. However, this treatment plan typically involves multiple interventions, a long and multidisciplinary process whereby placement of dental implants does not always result in prosthetic rehabilitation. Ritschl et al. reported a total of 63% of implanted patients receiving a prosthetic restoration [5], and Wetzels et al. reported functioning dental implant-supported prostheses in 73.9% of patients [6]. As most oncological patients who require fibula free flap reconstruction also undergo adjuvant radiochemotherapy due to malignant bone invasion, achieving oral rehabilitation for these patients is challenging due to the added xerostomia, trismus, and risk of osteoradionecrosis [7]. Moreover, this patient population has a poor prognosis (Stage IV disease), as the average survival of patients with a T4 oral cavity tumor is 29% after 5 years [8]. The low survival probably contributes to the low number of patients not receiving prosthodontic rehabilitation despite the presence of dental implants.

The immediate loading of the implants with a prosthesis was a second milestone and led to the jaw-in-a-day concept first described in a trauma case reconstructed with a scapula free flap in 2007 by Iain Hutchison and later by Levine and colleagues in 2013 by using a fully digital workflow [9,10]. Since then, the jaw-in-a-day concept has been optimized, and many institutions have published their experiences and specific digital workflows [11,12].

Virtual surgical planning (VSP) for the reconstruction of oncological defects has evolved significantly in recent years and has become indispensable due to its numerous advantages. However, some challenges have emerged that were not initially highlighted, such as difficulty in accurately estimating soft tissue thickness. Soft tissue interference can lead to translational and/or rotational shifts during the positioning of the fibula segments compared to the initial planning. Biologically, fibula segments can accommodate rotational discrepancies to some extent and still ossify reliably, but for applications involving 3D-printed reconstruction plates and the positioning of dental implants for immediate prosthetic rehabilitation, minimal shifts can impact the postoperative prosthetic outcome. By using a stackable system along with a manually prebent reconstruction plate, these minor movements due to soft tissue mass can be accommodated.

The purpose of this technical note is to present an innovative modular, stackable guide system for mandibular reconstruction with an osseous free flap, primary implantation with dental implants, and immediate oral rehabilitation with a provisory dental prosthesis at the point of care without the need for patient-specific titanium implants.

## 2. Materials and Methods

### 2.1. Patient Presentation and Preparation

Squamous cell carcinoma of the left floor of the mouth with clinical and radiographic mandibular invasion (cT4N2bM0) was diagnosed in a 69-year-old female patient with no relevant medical history. The patient had a normal mouth opening, dentition from 35 to 46 with pontics at 31 and 41, and a stable and reproducible occlusion. After multidisciplinary oncological consultation, the following treatment plan was agreed upon:

Tracheotomy; oncological resection with >1.5 cm resection margins; left unilateral radical neck dissection levels I–V; osteomyocutaneous reconstruction with a right fibular free flap; primary implantation with dental implants and immediate loading of the latter with an implant retained prosthesis.

Prior to surgery, the patient was evaluated to assess the vertical dimensions, smile line, dental occlusion, prosthetic space, and sagittal, transversal, and vertical jaw position. Preoperative data collection with a high-resolution (0.6 mm slice thickness) computed tomography (CT) scan (Somatom, Siemens, Munich, Germany) of the head and neck with the temporomandibular joint in centric relation and a CT angiography of the lower legs was performed. An intra-oral scan (Shining 3d, Stuttgart, Germany) of the upper and lower dental arches combined with an occlusion scan was collected.

### 2.2. Virtual Surgical Planning and 3D Printing

Virtual surgical planning (VSP) was performed by the surgeons and the prosthodontist, as there is no clinical engineer in the hospital. Detailed 3D models of the cranium and fibula were created from the CT scans using the Materialise Innovation Suite version 26 software (Materialise NV, Leuven, Belgium). Mandibular osteotomies were marked according to the Brown classification class IV (both canines and one angle) [13], followed by the incorporation of the fibula in the remaining mandible using the Enlight^®^ 2.0 reconstruction planner (Materialise NV, Leuven, Belgium). Careful positioning of the fibula parts is essential for optimal dental implant positions and sufficient height for the design of the prosthesis and postoperative cleaning of the implants. By using prosthetic-driven backward planning, the fibula parts were positioned and rotated based on the final position of the provisory dental prosthesis, in this case, based on the current position of the teeth [14] (Figure 1).

The 3D models of the cranium, mandible, neomandible, fibula, and dental arches were imported into 3-matic^®^ 18.0 (Materialise NV, Leuven, Belgium). The neomandible and dental arches were aligned on the original mandible using point-by-point registration. In this case, the current dentition of the patient served as the reference for the tooth setup of the provisional prosthesis. The dental implants were positioned based on the current dentition of the patient and were visualized with cylinders of 5 mm diameter. Customized cutting guides for the mandibular osteotomies (Figure 2A,B); a 3D model with a sleeve for vertical support of the fibula parts (Figure 2C), and a 3D model with the fibula parts in situ for the prebending of the 2.3 mm stock reconstruction plate (KLS Martin, Tuttlingen, Germany) (Figure 2D) were designed.

For the design of the fibula cutting guide, several requirements were defined:Presence of cutting guides for the fibula osteotomies to create the fibula segments in the correct dimensions (flanges were preferred);Space for placement of the prebent reconstruction plate;Control mechanism for the exact placement of the fibula parts in the mandible;Mechanism for accurate placement of dental implants;Mechanism for fixation of the prosthesis to the implants in the correct sagittal, vertical, and transversal positions;Creation of the complete construct on the lower leg with the vascular supply still intact to minimize ischemia time;Efficient workflow to minimize operating time.

To comply with the abovementioned demands, a modular, stackable guide system was developed based on the STAR concept used in dental implantology [15]. The flanges for the fibula osteotomies—holes for screw fixation to the fibula and connecting cylinders—were created in 3-Matic^®^ 18.0 (Figure 2F). These cylinders are designed in such a way that they can be removed during surgery to allow space for placement of the reconstruction plate (see further for more details).

All parts were duplicated and translated to the mandible using point-by-point registration. The guide system was then imported in Blender v3.6 with the Blenderfordental add-on (Blenderfordental Pty, Robina, Queensland, Australia) where the base plate with 3 attachment cylinders was designed. The Blenderfordental add-on was chosen for easy dental design of the provisional prosthesis and connecting arms. For accurate positioning of the different stackable guides, connecting arms were added to the fibula cutting guide, the implant guide, and the prosthesis positioning guide. These connecting arms fit exactly on the attachment cylinders (Figure 3).

These parts were again imported into 3-Matic^®^ 18.0, matched to the existing planning on the mandible and the fibula using point-by-point registration (Figure 4). An offset of 0.2 mm was used, and all objects were wrapped and checked for errors using the fix wizard. All guides were 3D printed in Ultrasint**^®^** polyamide 11 (Forward AM Technologies, Heidelberg, Germany) with a Farsoon SS403P 3D printer (Farsoon Europe, Stuttgart, Germany). The provisional dental prosthesis was 3D printed in polymer (Ultraprint Dental Denture UV, Heygears, Guangzhou, China—C&B Microfilled Hybrid, Nextdent, Soesterberg, Netherlands) with a Heygears A3D 3D printer (Heygears, Guangzhou, China).

### 2.3. Surgery

During the resection of the tumor, the 2.3 mm reconstruction plate was intraoperatively contoured on the 3D-printed neomandible (Figure 2D and Figure 5A). The plate was positioned just below the superior surface parts of the fibula guide. The bending of the plate started at the level of the left condyle (Figure 5B) and, by using bending pliers, was gradually adjusted towards the symphysis to achieve the desired final shape (Figure 5C). After the removal of the inferior portion of the fibula guide (Figure 6), the reconstruction plate could be fixated on the fibula segments (Figure 5D).

The osteomyocutaneous fibula free flap was prepared at the right leg in a supraperiosteal plane with a mobile skin island (4 × 3 cm) and a part of the musculus flexor hallucis longus (3 × 3 cm). The fibula guide was positioned and fixed with 9–11 mm long 1.5 mm screws through the fixation holes in the fibula guide (KLS Martin, Tuttlingen, Germany) (Figure 6A). Excessive deperiostation of the fibula segments was avoided by using piezosurgery (Piezosurgery Flex, Mectron spa, Carasco, Italy) for the osteotomies. The fibula segments not needed for the reconstruction were deperiostated and removed. The small connecting cylinders bridging the superior and inferior parts of the fibula guide were cut with a wire cutter, and the lower part of the fibula guide was removed, hereby mobilizing the fibula segments (Figure 6B,C). The fibula segments were positioned in the base plate guide with the connection arms fixed on the attachment cylinders (Figure 6D).

The prebent 2.3 mm reconstruction plate (KLS Martin, Tuttlingen, Germany) was positioned on the fibula segments and the 3D model and fixated to the fibula segments with 2.0 mm locking and non-locking screws securing the position of the fibula segments as preoperatively planned. Care was taken not to damage the contralateral mental nerve by bending the reconstruction plate around the mental foramen (Figure 5). After placement of the reconstruction plate, the 1.5 mm screws and the superior part of the fibula guide were removed from the fibula bone. The stackable implant guide was now positioned on the attachment cylinders of the base plate and the remaining dentition on the mandible model (Figure 6E). Three Bredent classic SKY 4.0 × 10 mm implants (REF: kSKY4010, Bredent medical, Senden, Germany) were placed on positions 41, 32, and 35 in the fibula bone based on the preoperatively indicated positions. Adequate primary stability (>35 Ncm) was achieved. Straight 3 mm unicone abutments (Ref: SKYUC003, Bredent medical, Senden, Germany) and polymer temporary cylinders (Ref: SKYUCPKK, Bredent medical, Senden, Germany) were positioned on the implants. The stackable implant guide was removed and the stackable prosthetic guide with the prosthesis in place was positioned on the attachment cylinders and the remaining dentition of the mandible model (Figure 6F). The provisional prosthesis was then picked up by fixing the polymer cylinders to the prosthesis with G-aenial^TM^ Universal Flo (GC Europe NV, Leuven, Belgium). The provisional prosthesis was removed from the abutments, and final prosthetic adjustments and finishing were performed by the prosthodontist. At the same time, the pedicle was prepared for sectioning, and the receptor site was prepared for transfer of the construct.

Finally, the prosthesis was placed back on the abutments, the pedicle was sectioned, and the fibula construct was transferred to the mouth (Figure 7). The fibula construct was manually placed in occlusion by the surgeons and fixed to the mandible with a combination of 2.0 mm locking and non-locking screws. During the transfer of the fibula complex to the oral cavity, the remaining dentate right mandible was placed manually in occlusion. The technique was comparable to the conventional approaches used for parasymphyseal and angular mandibular fractures. The fibula construct was first manually positioned and brought into occlusion. Once adequate occlusion and bone contact were confirmed medially, two bicortical screws were placed in the native right mandible for anchorage. These screws were not fully tightened to allow for slight movement. In the next step, the native mandibular ramus was secured with two non-locking screws, anchoring the entire fibula construct to the native mandible. After re-evaluation of the occlusion, the fibula was fully secured to the native mandible using multiple bicortical screws. The advantage of using non-locking screws and initially leaving them slightly loose is that it allows for the detection of any tension in the neomandibular reconstruction. By gradually tightening the screws in a sequential manner, any potential tension can be relieved.

The peroneal vessels were anastomosed to the right arteria thyroidea (end-to-end) and the right vena jugularis externa (end-to-side). The total ischemia time was 2 h and 19 min. The flexor hallucis longus muscle flap was draped and sutured around the implants. The remaining mucosa and the skin island were sutured for soft tissue coverage. The skin island was positioned posteriorly and was used for the monitoring of the free flap.

The donor site was closed primarily in layers, and a small area was covered with a split-thickness skin graft from the right calf.

### 2.4. Peri- and Postoperative Management

The total duration of the surgery was 9 h and 35 min. The patient remained sedated during the first night in the intensive care unit. Postoperatively, the patient received intravenous antibiotic prophylaxis with amoxicillin clavulanic acid, 1 g four times daily for 7 days, and subcutaneous enoxaparin, 60 mg twice daily. Nasogastric tube feeding was initiated with 1.5 L of Nutrison Protein Plus MF at a rate of 60 CC and was gradually increased under the supervision of the dietitian.

After five days, the patient was transferred to the inpatient clinic, and physiotherapy and gait rehabilitation were initiated. The tracheostomy tube was successfully removed on postoperative day 15, after which speech therapy was intensified, focusing on optimizing oral, pharyngeal, and laryngeal mobility. After 23 days, the patient was discharged, continuing nasogastric tube feeding at home consisting of 1 L of Nutrison Protein Plus Multi Fibre overnight and 500 mL of Nutrison Protein Plus Multi Fibre during the day at 85 CC. However, her hospitalization was marked by persistent dysphagia for both liquids and spoonable foods. Considering the need for adjuvant treatment with radiochemotherapy, which could potentially further impair swallowing function, the decision was made to place a percutaneous endoscopic gastrotomy (PEG) tube on an outpatient basis 4 weeks after the surgery.

Adjuvant radiochemotherapy was initiated 5 weeks postoperatively and continued for 6 weeks. A total final dose of radiation was achieved in 33 fractions of 2 Gray (Gy), amounting to 66 Gy. Concurrently, chemotherapy with cisplatin 40 mg/m^2^ was started weekly, but the last cycle was not administered due to increased hearing loss.

## 3. Results

The provisional prosthesis provided a stable and balanced occlusion. Six weeks postoperative, spontaneous mucosalisation of the gracilis muscle was seen, providing a thin and immobile peri-implant soft tissue bed around the dental implants. Combined with sufficient space for implant cleaning, there were no periodontal pockets and only minimal bleeding on probing. The patient had a good temporomandibular joint function with a mouth opening of 25 mm 2 months postoperatively (Figure 8A,B), which remained stable 9 months after surgery (Figure 8C, D). Following a 9-month healing period, to allow complete osseointegration and soft tissue maturation, the final prosthesis was fabricated using conventional impression techniques and bite registration. Due to the presence of adequate vertical dimension, a CAD/CAM milled bar with locator abutments can be fabricated to support the final removable prosthesis. By using this type of retention system, the locator abutments can be designed to be perfectly parallel to each other. This will minimize abrasion of the abutment, potentially reducing the frequency of the patrix and matrices replacement [16,17]. This type of rehabilitation also provides adequate lip support, masticatory force, and retention while still allowing for good oral hygiene and surveillance of the reconstructed site.

In the attached video, the patient speaks in Dutch about how she is experiencing her new situation, thereby emphasizing that she can speak very well and is able to eat almost all types of food. In the end, the dynamic function of the temporomandibular joint is demonstrated (Appendix A).

A postoperative comparison of the preoperative planning and the postoperative segmented mandible resulted in a mean difference of 0.7 mm (range: −1.9940 mm; 5.4144 mm) (Figure 9).

## 4. Discussion

The aim of this technical note is to present a novel technique for jaw-in-a-day reconstruction. A modular, stackable guide system for mandibular reconstruction with osseous free flaps combined with primary implantation and immediate oral rehabilitation without the need for patient-specific titanium implants was developed. This stackable guide system is designed and fabricated at the point of care, therefore potentially resulting in faster treatment times and reduced costs.

There are several studies describing specific workflows for successfully performing jaw-in-a-day reconstruction. The use of stackable guides has not been reported until now. By uncoupling the positioning of the fibula segments with the dental implant placement, small inaccuracies during surgery can be mediated.

Compared to other studies, this technique aims to be cost-effective by not outsourcing virtual surgical planning and by not using 3D-printed patient-specific titanium implants. The use of these tools has proven to be accurate but remains expensive [18]. By positioning the fibula construct based on the dental occlusion, there is also no need for a ‘pick-up’ of the prosthesis, an intraoperative intra-oral scan, or positioning of the prosthesis after fixation of the bony segments [19,20,21]. This prosthetic-driven approach reduces the number of steps necessary to achieve prosthetic rehabilitation.

By using computer-assisted design and computer-aided manufacturing, it is possible to achieve predictable and accurate results for complex surgeries. As there is no clinical engineer at our hospital, the VSP and CAD/CAM are performed in-house by oral- and craniomaxillofacial surgeons. By using simple and efficient software programs, such as the Enlight^®^ 2.0 reconstruction module, the updated 3-Matic^®^ 18.0 with fast workflows for flange design and connecting bridges, and the Blenderfordental add-on, VSP has become easier, faster, and more feasible for surgeons. By using open-source software programs, the VSP can be exported and imported into different software packages, benefitting from each of their strong points. In this case, the stackable connections were designed in Blender with the Blenderfordental add-on. VSP, performed by surgeons, offers the unique advantage of understanding both the clinical and technological possibilities and limitations, resulting in creative designs and immediate feedback during surgery. The biggest disadvantage is the time needed for planning these complex cases. However, with the rapid evolution of artificial intelligence and simplifications of the current software packages, it is highly likely that VSP of complex surgeries will become less time-consuming. As the entire design and production workflow can be performed in-house and there is no need for patient-specific implants, this method could prove to be cost-effective, as well. More research to confirm this assumption is necessary.

A major limitation of VSP for mandibular reconstruction is the surgeon’s inability to accurately estimate the thickness of the soft tissues (in this case, the muscles and skin around the fibula). The volume of these soft tissues can cause slight, unexpected translational, or rotational shifts during the positioning of the fibula segments. Using a stackable guide system helps to solve this issue by decoupling the implant’s placement from the precise position of the fibula segments. The stackable system, together with the manually adjusted reconstruction plate, allows us to accommodate these minor rotational or translational movements in the fibula segments. This ensures that the complete prosthetic construction can be accurately built, as the stackable system indicates the implant positions independently from the fibula segments’ placement.

A manually adapted reconstruction plate provides an important advantage, namely its adaptability, over a 3D-printed reconstruction plate, which is often rigid and, therefore, difficult to adapt during surgery. The placement of the screws is hereby also subjected to more degrees of freedom.

Oncological osseous resection and reconstruction are usually preserved for end-stage oncologic disease. As the prognosis of these patients is poor, it is important to rehabilitate the patient as soon as possible. By using the jaw-in-a-day concept, the patient can be anatomically reconstructed and functionally rehabilitated in a one-stage surgery. As these patients are usually treated with adjuvant radiochemotherapy, the resulting trismus, xerostomia, and negative effects on the vascularization complicate subsequent interventions [22].

It is generally accepted that the primary placement of implants during ablative surgery results in lower individual costs and a higher number of functional mandibular prostheses after the completion of oncological therapy [23,24]. Alberga et al. [25] conclude that primary implant placement in the native mandible should be considered the gold standard within head and neck oncology.

By immediately placing the dental prosthesis on the implants, the number of additional surgeries can be reduced, and the patient can immediately start orofacial revalidation, improving their quality of life tremendously. As shown in oral implantology, the immediate loading protocol can be applied in a standardized manner by using 3D-printed stackable guides [15,26]. The advantages of working with stackable guides include reduced operating time, higher precision, reproducibility, and respect for the prosthetic plan [27].

The jaw-in-a-day concept requires a multidisciplinary approach. In this case, the maxillofacial surgeons were responsible for the VSP and CAD/CAM of the guides, the prosthodontist for the fabrication and placement of the provisional and final prosthesis, the ear, nose, and throat surgeon for neck dissection and tracheotomy, and the plastic surgeon for the fibula free flap and vascular anastomosis. Due to this multidisciplinary approach, several steps in the surgery can be performed simultaneously, resulting in an operating time of 9 h and 35 min. Close cooperation between the surgeons and the prosthodontist is essential in this process [28]. Preoperatively, a thorough clinical and radiological analysis of the patient’s dental and maxillofacial status, with specific attention given to dental occlusion, bite height, skeletal relationships, and patient wishes, is essential for a good outcome. For virtual surgical planning, data collection through clinical photos, intra-oral scans, and high-resolution CT imaging is necessary for accurate planning and proper fitting of the created guides. For the provisional prosthesis, a screw-retained bridge on unicone abutments is preferred, but after healing, it is replaced with an implant-supported overdenture. These are considered a favorable treatment option for patients who have undergone extensive reconstruction from oral cancer. While the removability of the prosthesis highly enhances patient oral hygiene and facilitates further monitoring of the reconstructed site, this treatment modality requires sufficient vertical and horizontal space, depending on the prosthesis type [29,30]. Excessive vertical space, however, may lead to long abutments and an increased implant/crown ratio, which can result in early implant failure [31]. Therefore, the position of the fibula parts in the reconstructed mandible should be discussed with the prosthodontist.

To achieve a balanced occlusion in a correct vertical dimension with acceptable facial harmony, VSP, for this concept, is based on backward planning. In this case the patient had sufficient remaining dental elements with a good occlusion and good vertical dimension. If the patient is edentate, a dental prosthesis can be created to determine the vertical dimensions and positions of the teeth. The preoperative high-resolution CT scan should be taken with the dental prosthesis in place, preferably marked with scan markers to allow accurate alignment during VSP. By using a dual-scan protocol, the prosthesis can be matched virtually [32].

The fibula parts are positioned more than 15 mm below the occlusion line to provide sufficient space for oral hygiene [15,23,24,25,26,27]. A part of the flexor hallucis longus was elevated and draped around the implants for soft tissue healing, minimizing the soft tissue thickness for good peri-implant conditions.

This modular, stackable guide system shows great potential to achieve immediate oral rehabilitation in an efficient, accurate, and reproducible manner. However, certain aspects could be improved:The size of the base plate interfered with the lower leg, the skin island, and the muscle flap during surgery;The supporting sleeve for vertical support of the fibula parts resulted in a higher position of the fibula parts due to soft tissue interference;The placement of the dental implants was difficult because there were limited reference points and no control of drill direction;Because the location of the implants was not exact, the placement of the provisional dental prosthesis was time-consuming;The provisional dental prosthesis lacked vertical support for efficient positioning.

The authors believe that this technique can also be extrapolated to other bony defects in the skull, such as Brown class III, IV, or maxillectomies. In future cases, the base plate must become smaller, and a fully guided dental implantation protocol should be added to this guide system. This should result in reduced operating times and a more accurate result.

## 5. Conclusions

This novel, innovative stackable guide system uncouples the positioning of the osseous segments and the dental implant placement, offering the advantages of virtual surgical planning but still allowing the possibility for corrections of intraoperative inaccuracies. The design and production take place at the point of care and potentially result in a fast and cost-effective result. More research to confirm the usability of this new technique is necessary.

## Figures and Tables

**Figure 1 bioengineering-11-01254-f001:**
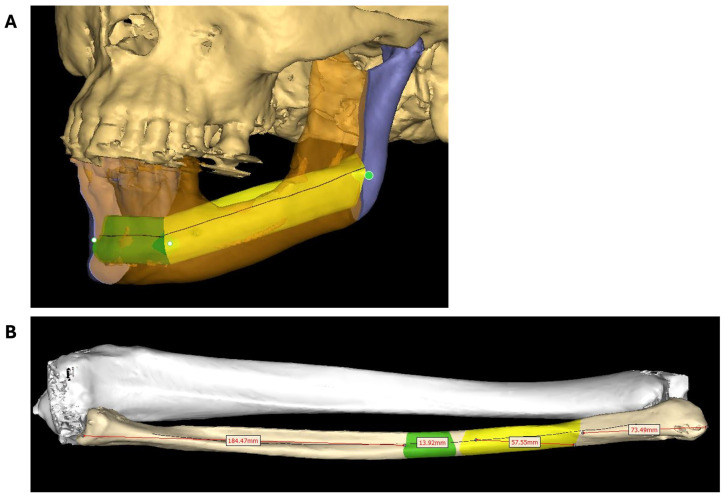
VSP in Enlight^®^ 2.0 (Materialise, Leuven, Belgium). (**A**) Position of the two fibula parts in between the remaining mandible after virtual resection of the tumor. The height and rotation of the fibula parts are essential parameters to allow optimal implant placement. Purple: remaining mandible parts; translucent orange: resected mandible; green + yellow: two-piece fibula for mandibular reconstruction. (**B**) Visualization of the right tibia and fibula bones. In yellow and green, the corresponding fibula parts are visible. The measurements of each part and the distance to the lateral malleolus and fibula head are visualized (red text boxes).

**Figure 2 bioengineering-11-01254-f002:**
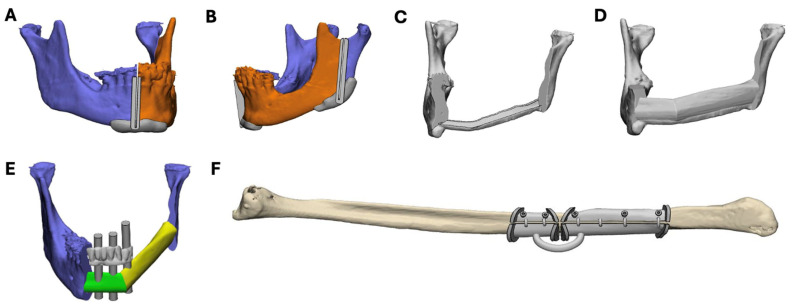
VSP of cutting guides and templates designed in 3-Matic^®^ 18.0 (Materialise, Leuven, Belgium). (**A**,**B**) Cutting guides for the right and left mandibular osteotomy. (**C**) Positioning model with vertical support for positioning of the fibula segments. (**D**) Neomandible for prebending of the reconstruction plate. (**E**) Neomandible and intra-oral scan of the dentition for height estimation of the fibula segments. The dental implant position is visualized with 5 mm diameter cylinders. (**F**) Initial cutting guide for fibula osteotomies.

**Figure 3 bioengineering-11-01254-f003:**
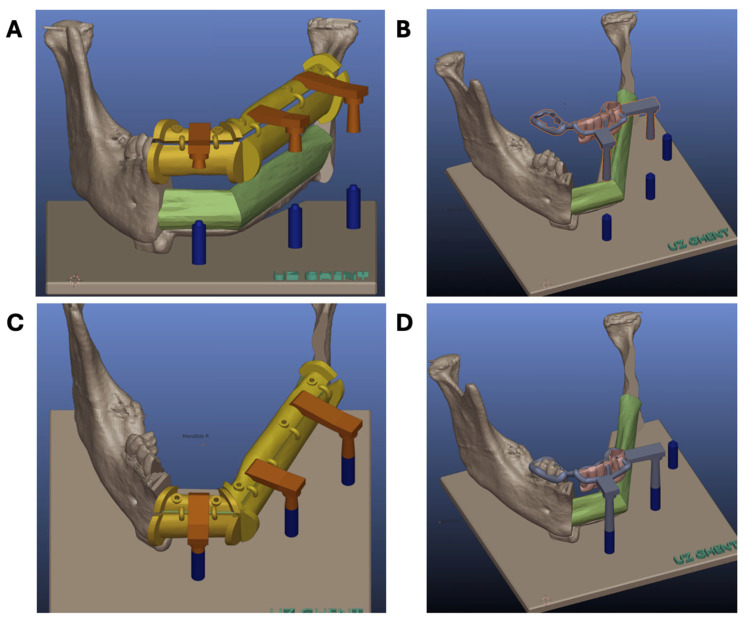
Design of the base plate, attachment cylinders (blue), and connecting arms (brown) in Blender v3.6 with the Blenderfordental add-on. (**A**,**C**) Connecting 3 attachment cylinders with three connecting arms to position the fibula guide and fibula parts in a correct position. (**B**,**D**) Connecting two attachment cylinders with two connecting arms and an extension towards the native dentition to position the provisional prosthesis.

**Figure 4 bioengineering-11-01254-f004:**
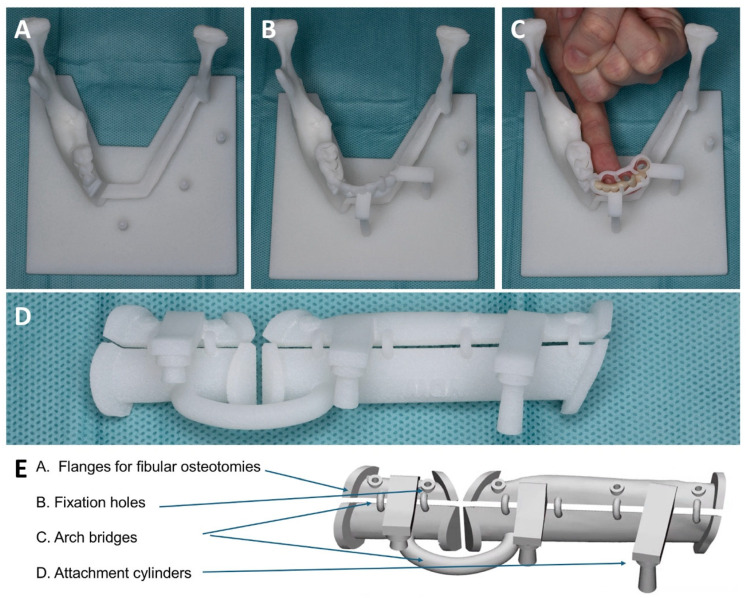
Modular, stackable guide system. (**A**) Base plate with three attachment cylinders to connect the fibula cutting guide for optimal positioning of the fibula segments. The same attachment cylinders are used for positioning the stackable guides for implant placement and prosthesis positioning. A supporting sleeve, bridging the remaining native mandibular parts, allows correct vertical positioning of the fibula segments. (**B**) Base plate with stackable implant drill guide connected to the attachment cylinders and to the right mandibular teeth using a dental extension. This guide holds occlusal drill holes for positioning the dental implants in the fibula segments. (**C**) Base plate with stackable prosthesis guide (white) and the provisional prosthesis positioned in the correct vertical dimension. (**D**,**E**) Fibula guide with sufficient contact surface to provide accurate positioning on the fibula; fixation holes for fixation to the fibula with 1.5 mm screws; 2 mm arch bridges to connect both superior and inferior surface areas; 5 mm arch bridge connecting the two fibula guide parts; flanges serving as cutting guides; connection arms for positioning of the fibula segments on the base plate.

**Figure 5 bioengineering-11-01254-f005:**
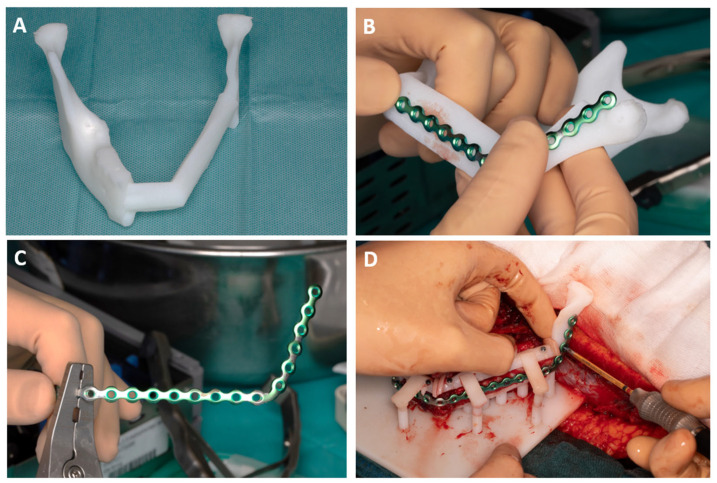
Prebending of the 2.3 KLS Martin reconstruction plate: (**A**) A 3D model of the neomandible. (**B**) The reconstruction plate is placed on the neomandible model, starting from the condyle. (**C**) The plate is adjusted using a surgical marker and folding pliers. (**D**) The prebent reconstruction plate is fixed to the fibula segments that are positioned by the superior surface parts of the fibula guide.

**Figure 6 bioengineering-11-01254-f006:**
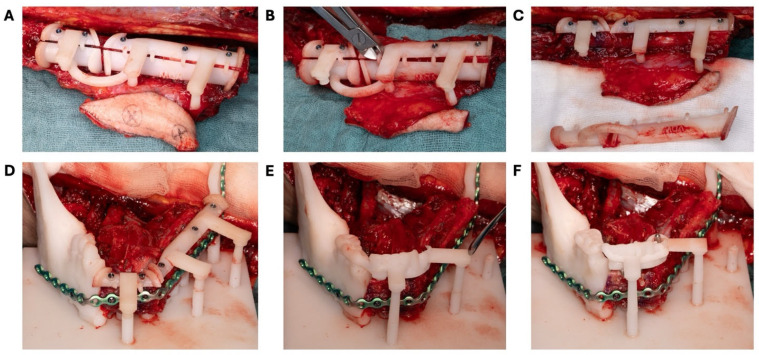
Surgical steps for guide positioning. (**A**) The fibula guide was fixed on the right fibula with 1.5 mm screws. Skin island attached to the fibula. (**B**,**C**) After performing the fibula osteotomies, the 2 mm arch bars were cut to remove the inferior parts of the fibula guide. (**D**) Positioning of the fibula parts by connecting the connection arms of the fibula guide to the attachment cylinders of the base plate. Fixation of the prebent 2.3 mm KLS Martin reconstruction plate with locking and non-locking screws. (**E**) After the removal of the superior part of the fibula guide, the stackable implant guide for dental implant placement was placed. An extension was made for support of the remaining dentition for stability reasons. (**F**) Placement of the stackable prosthetic guide to check implant position before connecting the provisional prosthesis.

**Figure 7 bioengineering-11-01254-f007:**
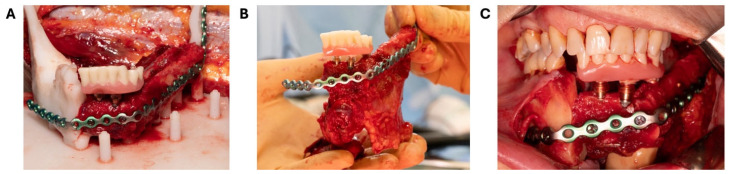
Incorporation of the fibula construct. (**A**) Neomandible with fibula segments positioned and fixed to the reconstruction plate. Temporary prosthesis fixed to three Bredent classic SKY 4.0 × 10 mm dental implants. The fibula construct is still attached to the pedicle. (**B**) Fibula construct after sectioning of the pedicle with attached skin island and flexor hallucis longus muscle flap for transfer to the patient’s mouth. (**C**) The fibula construct is positioned in occlusion and fixed with locking and non-locking screws to the mandible.

**Figure 8 bioengineering-11-01254-f008:**
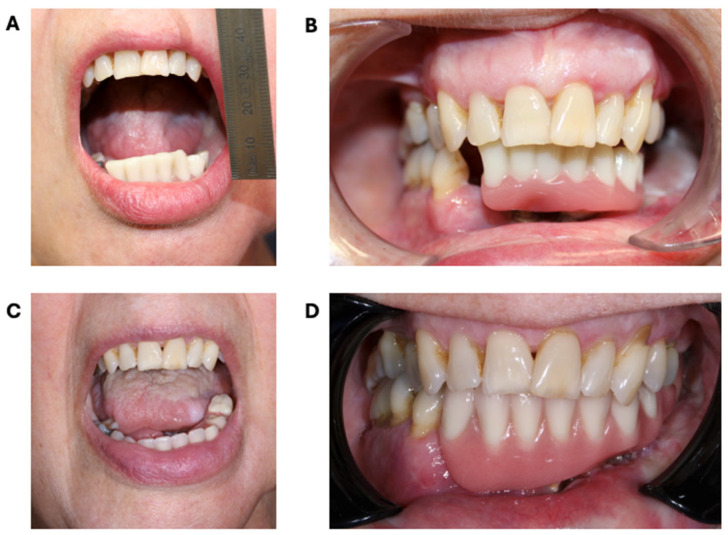
Postoperative clinical images 2 (**A**,**B**) and 9 (**C**,**D**) months after surgery. (**A**) Good temporomandibular function with interincisal mouth opening of approximately 25 mm. (**B**) Provisional dental prosthesis fixed on dental implants with stable occlusion. (**C**) Stable temporomandibular function after radiochemotherapy. (**D**) Stable occlusion with the final prosthesis in place.

**Figure 9 bioengineering-11-01254-f009:**
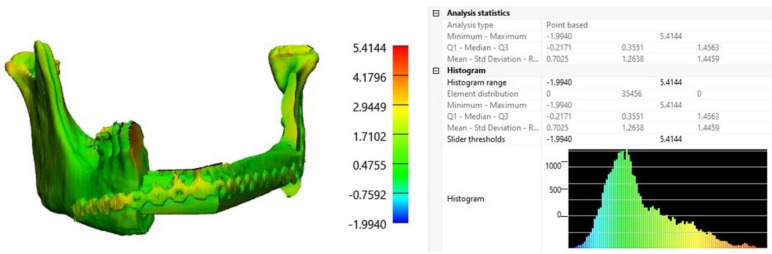
Superimposition of preoperative planning and postoperative result. Mean difference of 0.70 mm (range: −1.9940 mm; 5.4144 mm).

## Data Availability

The original contributions presented in this study are included in the article; further inquiries can be directed to the corresponding author.

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
