# Peer review of "Mandibular Reconstruction with Osseous Free Flap and Immediate Prosthetic Rehabilitation (Jaw-in-a-Day): In-House Manufactured Innovative Modular Stackable Guide System"

_bioengineering, 2024, doi:10.3390/bioengineering11121254_

Round 1

Reviewer 1 Report (Previous Reviewer 2)

Comments and Suggestions for Authors

The authors have addressed all the raised queries thoroughly, and the overall quality of the manuscript has significantly improved. 

Reviewer 2 Report (Previous Reviewer 1)

Comments and Suggestions for Authors

Overall a clear presentation of an modification a "jaw-in-a-day" concept. This "jaw-in-a-day" surgery especially with the base plate is in an interesting innovation of the surgery. This paper now provides more detail in image and videos which allows better understanding of the work-flow. Generally a clear and well written technical note article.

This manuscript is a resubmission of an earlier submission. The following is a list of the peer review reports and author responses from that submission.

Round 1

Reviewer 1 Report

Comments and Suggestions for Authors

Jaw in a day is not a new concept, but the authors described a new idea using stackable design  with attachment cylinders of the base plate as a guide. The authors did highlight the benefits and disadvantage of this method and highlighted the steps to design such guide.

The authors should also clarify regarding the reconstruction plate, where it is mentioned about pre-bending it, but how was it positioned accurately as in the planning? The exact position to drill the screw hole and angulation of the plates are not clarified and should be added. This is the final piece as all the implant  / prosthesis / osteotomy has been guided extra-orally but how did they transfer this fibula-prosthesis complex accurately into the surgical field.

Author Response

Comment 1: Jaw in a day is not a new concept, but the authors described a new idea using stackable design with attachment cylinders of the base plate as a guide. The authors did highlight the benefits and disadvantage of this method and highlighted the steps to design such guide.

Answer: Thank you for reviewing our article and your positive feedback.

Comment 2: The authors should also clarify regarding the reconstruction plate, where it is mentioned about pre-bending it, but how was it positioned accurately as in the planning?

Answer: Thank you for this valuable comment. We have added an extra figure (Figure 5) and explained the process of how the reconstruction plate was bended. As the superior surfaces of the fibula guide remain attached to the fibula segments it is important to place the plate at the level of the removed inferior surface areas. This was explained in the added text (lines 339 - 345).
We also added a paragraph explaining how the fibula construct is accurately positioned in the mouth (lines 481-495). 

Comment 3: The exact position to drill the screw hole and angulation of the plates are not clarified and should be added. This is the final piece as all the implant  / prosthesis / osteotomy has been guided extra-orally but how did they transfer this fibula-prosthesis complex accurately into the surgical field.

Answer: We have added a paragraph in the discussion explaining why we like to accept some freedom in the placement of the osteosynthesis plate. There are no predrilled holes for placement of the reconstruction plate. The plate is attached to the mandible based on the dental occlusion of the patient (Lines 785 - 798).

We hope these changes will be sufficient for accepting our article for publication. Please do not hesitate to provide additional comments to improve our work.

Kind regards,
Matthias Ureel

Reviewer 2 Report

Comments and Suggestions for Authors

The case report/technical note entitled “Mandibular reconstruction with osseous free flap and immediate prosthetic rehabilitation (jaw-in-a-day): In-house manufactured innovative modular stackable guide system” by Matthias Ureel et al. contains interesting results. It can be accepted for publication once the authors properly address the following issues, in the order they appear in the manuscript:

1. The Introduction section is disproportionately short compared to the other sections, and is supported only by a few references. The authors should highlight the novelty of their approach by discussing its unique advantages and potential drawbacks compared to prior studies. Thus, it should be clearer to the reader why this approach is superior to the others reported so far.  

2. All presented figures must include magnification bars.

3. This Reviewer recommends moving the paragraphs on page 7, lines 161 to 201, page 8, lines 219 to 230, and page 9, lines 231 to 239 to the “Materials and Methods” section. To improve organization and accessibility, the authors might also consider dividing this section into subsections.

4. The “Discussion” section would benefit from a more in-depth comparison of the authors' findings with relevant studies, with a focus on the unique advantages their approach offers to the field.  

5. This case report/technical note lacks a “Conclusions” section. This section should succinctly summarize the main findings and implications, offering valuable insights for readers and emphasizing significant contributions to the field.

6. Can this innovative modular stackable guide system be applied to other anatomical regions beyond mandibular reconstruction? This information is important and should be included in the revised version of the manuscript.  

7. Ref. 6 is incomplete and needs revision.

Author Response

Thank you for your valuable comments. We have changed the manuscript accordingly and hope these changes will be sufficient for publication. Please see the attachment.
